# Reconstruction of missing data in transferred generative adversarial networks with small sample data

Jing He[1], Hongrun Chen[2]*, Zhenwen Sheng[1]

1 College of Engineering, Shandong Xiehe University, Shandong, China, 2 China Railway Engineering Equipment Group Co. Ltd, Zhengzhou, Henan,China

* hongrunchen@sdxiehe.edu.cn

## Abstract

Under special working conditions, data collection systems of heavy-duty trains may be faced with a small sample size and missing data when executing measurement, operation, or maintenance tasks. Existing generative modeling methods are ineffective in reconstructing missing data with such a small sample. Hence, we set up a frame of migration learning generative adversarial network for small data samples, in which a new variational autoencoder semantic fusion generative adversarial network (VAE-FGAN) is developed to reconstruct missing data. First a GRU module is introduced in the encoder to fuse the underlying features of the data with higher-level features, which enables the VAE-FGAN to learn the correlation between the measured data through unsupervised training. Second, an SE-NET attention mechanism is introduced into the whole generative network to enhance the expression of the feature extraction network on data features. Finally, parameters are shared through migration learning and pre-training, thereby eliminating the difficulty in training the model due to the small size of certain operation and maintenance data. Experimental results show that the reconstruction accuracy indices MAE and MAPE can be kept below 1.5 when measured data is missing; the reconstructed data also fits well to the distribution trend of the measured data.

## 1. Introduction

Given the importance of safety in railway development, health management of railway systems has become a research hot spot [1–4]. As such, it is critical to collect high-quality operation and maintenance data of trains [5,6]. However, heavy-duty trains in China normally operate over a large range, and they are accompanied by many crossings and complex operating conditions, often running through a variety of complex environments such as rolling mountains and continuous tunnels. Such environments generate network failures, transmission interruptions, harmonic interference, and other phenomena, which result in a large amount of missing operation

**Data availability statement:** All data underlying the findings described has been uploaded as supplementary information.

**Funding:** The author(s) received no specific funding for this work.

**Competing interests:** The authors have declared that no competing interests exist.

and maintenance data. Lacking significant data jeopardizes the assessment of status and system faults of a train. Therefore, the missing data must be reconstructed to the maximum extent possible.

Some traditional missing data interpolation methods, such as the K-nearest neighbor (KNN) algorithm [7], expectation maximization (EM) algorithm [8], and others based on mathematical statistics and mechanisms for reconstructing missing data have low accuracy and poor results. Since various devices are involved in the operation and maintenance systems of heavy-duty trains, there are strong data correlations and complex data networks to consider. Therefore, it is difficult to effectively interpolate missing data with a pure mathematical model.

In the inspection of a train's measured data, the correlation between measured data obtained from different devices can be a basis for data reconstruction. With the development of deep learning, some technical approaches solve defective image restoration and image super-resolution reconstruction [9–11]. The reconstruction of missing data from train measurements is similar to the restoration of super-resolution images.

Goodfellow et al.[12] made a breakthrough by proposing the now very popular method of applying generative adversarial networks (GANs) to the reconstruction of missing data. Wang et al.[13] integrated the advantages of WAGN training and proposed a reconstruction method for missing data in power systems. Meanwhile, Liu et al.[14] proposed a generative adversarial network data reconstruction framework combining dual semantic sensing, and they effectively improved data reconstruction accuracy. Fan et al.[15] suggested a segmentation-based conditional generative adversarial network, which achieved better reconstruction results with seismic load sample data. The unique characteristics of the dataset discussed in this paper present a challenge in training GAN models with discrete data. As a result, we cannot guarantee that these models will produce samples that accurately represent the same distribution as the original dataset derived from random background noise. Additionally, they may not reliably reach Nash equilibrium, leading to potential gradient loss. This framework underscores the significant benefits of integrating a variational autoencoder into the model and generating an adversarial network.

For instance, Lin et al. [16] fused an encoder into a generative adversarial network and applied a gradient penalty to avoid pattern collapse. Moreover, Fang et al. [17] proposed a missing data reconstruction strategy based on generative adversarial networks. While these variants of the GAN model enhance accuracy as the number of model layers increases, they also impose higher demands on the dataset and necessitate a more significant amount of data for practical model training. As a result, their efficacy is limited when applied to small sample datasets.

Deep learning techniques rely on complete data to train deep network structures. Although train records contain much repetitive operation data, such data is still far too small for fault feature learning. Jiao et al.[18] constructed a dynamic migration learning mechanism to regulate classification criteria.This approach to transfer learning relies on transferring within homogeneous data, struggling to achieve effective

migration across different datasets.Bai et al. [19] proposed a cross-domain migration learning framework to address the problem of large differences between data samples. Additionally, Azzam et al. [20] applied the "cluster assumption" to constraints and mutually reinforced training to maximize the transfer of knowledge and achieve migratory associations between data from different domains.Some researchers have successfully applied transfer learning to generative models, yielding excellent results. For instance, Liang et al.[21] successfully migrated view labels from the source domain to the target domain, facilitating unsupervised cross-domain pedestrian recognition by embedding the label features into the generative adversarial networks.Migration learning ensures that features and parameters learned under different data are shared [22], thus effectively eliminating the difficulties in training deep network models with small samples of train operation and maintenance data.

In the present paper, we propose a variational autoencoder semantic fusion generative adversarial network (VAE-FGAN) for solving the reconstruction of missing train data. The key innovations of our work are listed below.

(1) For train application, we combine migration learning with VAE-FGAN networks to build a missing data reconstruction framework, which achieves parameter sharing through transfer learning and model pre-training to ease the difficulty in training text models due to the small amount of certain operation and maintenance data.

(2) A VAE is introduced instead of the generator of the original GAN, and the unique network advantages of VAE are put to good use to overcome the instability of the generation process caused by the adoption of random noise. A GRU module is introduced to the encoder to learn the deep feature distribution of the original data.

(3) An SE-NET attention mechanism is introduced into the generative network to enhance the expression of the feature extraction network on data features.

(4) Data interpolation visualization plots and the evaluation indices show that the reconstructed data can not only maintain high reconstruction accuracy but also be consistent with the measured data in terms of distribution patterns.

The remainder of this paper is as follows: Section 2 introduces related work.Section 3 outlines the overall design framework proposed herein and details the VAE-FGAN model. Section 4 evaluates the proposed method through various experiments. Section 4 presents the conclusions.

## 2. Related work

### 2.1. Generative Adversarial Networks

A GAN consists of two main components: a generator and a discriminator. The generator's role is to create synthetic data, while the discriminator's task is to differentiate between real and fake data. Through adversarial training, these two components engage in a continuous improvement process, with the generator enhancing the quality of the generated data to make them nearly indistinguishable from authentic data. The ultimate aim of GAN is for the generator to produce data so realistically that it successfully deceives the discriminator, who strives to identify accurate data from the generated samples accurately.

GANs have found significant applications in image generation and data enhancement. For instance, Zhou et al. introduced two GAN-based generative frameworks, COutfitGAN [23] and OutfitGAN [24]. Experimental results demonstrated superior performance to other leading methods regarding similarity, authenticity, and compatibility. Su et al. [25] developed a two-stage GAN-based image synthesis approach. This method first learns the universal knowledge across various classes and applies this shared knowledge to each class, integrating it with class-specific knowledge to reproduce high-quality images. In recent years, GANs have been used to reconstruct missing data. However, in the high-speed train project, the challenge of missing data from multiple sensors hampers the traditional generative adversarial network model's ability to enhance the Nash equilibrium and, therefore, cannot reconstruct the missing data accurately.

## 2.2  Current status of missing data reconstruction research

Missing data-filling techniques are mainly classified into traditional and deep learning-based approaches. Conventional methods are divided into three categories: probability-based [26], interpolation-based [27], and similarity-based [28] methods. The expectation maximization (EM) algorithm is a prominent example of probability-based methods. Mouret et al.[29] proposed an improved EM algorithm to manage potentially missing data and effectively address noise and interpolation issues within missing non-Gaussian data. While the EM algorithm is widely used in handling missing data, its computational speed decreases significantly as the missing data in the dataset increases. Interpolation-based methods, which include linear interpolation, spline interpolation, and polynomial interpolation, infer missing values from known data points. These techniques tend to deliver more accurate results when fewer missing values exist. However, their performance degrades when faced with many gaps in the data. Similarity-based methods usually include k-nearest neighbor (KNN), K-mean, and mean interpolation. The KNN algorithm [30] is computationally faster than EM algorithms, yet its accuracy decreases when missing extreme sample values. In summary, the traditional data reconstruction methods often fall short when addressing the challenges posed by missing data in high-speed train measurement data.

Deep learning-based missing data-filling methods can be categorized into three types based on training strategies: autoregressive, automatic encoding, and adversarial training. Autoregressive methods typically utilize recurrent neural networks (RNNs) or their variants to predict time steps of missing data using complete or interpolated time steps. This approach capitalizes on RNN's strengths in processing temporary data, although it may lack a global receptive field, resulting in potential error accumulation.

Automatic encoding methods, such as autoencoders, compress high-dimensional inputs into low-dimensional hidden states through encoders, followed by the reconstruction process through decoders. These encoders and decoders can be multilayer perceptrons, convolutional neural networks, or other types of neural networks. However, using bottleneck structures in autoencoders can lead to information loss. Adversarial training methods leveraged the interpolation strategy of GANs, unsupervised generative models. GANs can autonomously learn data distributions and features, enabling them to generate data conforming to them.

## 3.  Model framework

This section provides a comprehensive overview of the missing data reconstruction architecture utilized in the VAE-FGAN. We start with an overall description of the missing data reconstruction framework designed for generative adversarial networks, particularly with small sample data outlined in Section 3.1. Next, Section 3.2 delves into the VAE-GAN model design. Section 3.3 focuses on the development of the GRU module, while Section 3.4 elaborates on the SE-NET attention mechanism. Finally, Section 3.5 details the data reconstruction process.

## 3.1.  Missing data reconstruction framework using a Transfer GAN

When a train runs through continuous tunnels and rolling mountains, partial data samples of some features go missing. In addition, we must consider the special nature of the data recording of the data collection system for heavy-duty trains: i.e., the number of sensors on the train is limited; train operation data is not collected at every moment but only when data abnormality occurs. Thus, the data sample volume of abnormal state features is small, and hence it is difficult to conduct operation and maintenance diagnosis. To address this, we proposed a migration learning generative adversarial network framework for small data samples. We established a novel variational autoencoder fusion generative adversarial network (VAE-FGAN) to reconstruct missing data. As shown in Fig 1, the reconstruction framework of this paper includes the following three steps.

The first step in our process is data preprocessing. Convolutional neural networks are particularly adept at handling image data, yet the experimental data used here consist of time-varying, multi-dimensional operation and maintenance data. Anticipating future operation and maintenance data will tend to be high-dimensional, time-varying, and nonlinear; we

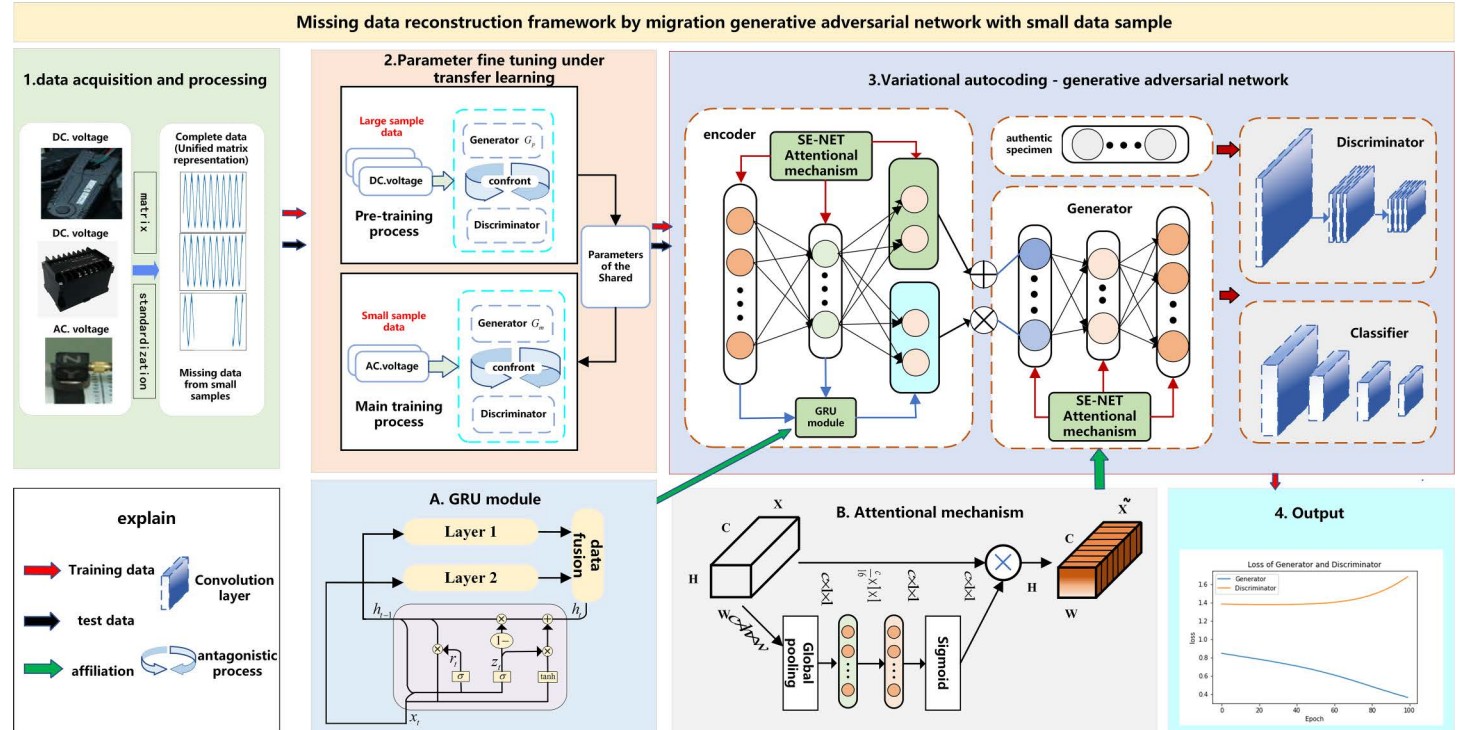

**Fig 1. Missing data reconstruction framework using variational autoencoder generative adversarial networks with a small data sample.**

transformed the data into a two-dimensional convolutional matrix format. This adaptation aids the model in locating the missing data and learning the contextual feature correlation between the missing and the existing data.

The second step involves leveraging a transfer learning model to facilitate parameter sharing between the two networks. In this process, the encoder and generator are pre-trained using sufficient data sample features. Subsequently, the trained parameters are migrated to the leading training network, which is fine-tuned using a small number of feature data samples. The migrated parameters encompass Adam optimizer parameters, loss function, and structural hyperparameters used in the encoder, generator, and discriminator networks.

The third step details the architecture of the VAE-FGAN network. We improved the generative adversarial network by integrating the VAE-GAN, the VAEs encoder E, generator G, and discriminator D networks. An innovative coded semantic fusion structure was created by combining the GRU module with the generator, known for its unique advantage in acquiring semantic information. This structure effectively extracts the semantic information from the upper and lower convolutional layers during the data training. Considering the influence of the deep features of the data on the generated data, we incorporated the SE-NET attention mechanism to encoder E and generator G. This enhancement aims to improve the feature learning and data generation capabilities on datasets with a small number of samples, enabling better reconstruction of the missing data of datasets. The transfer learning model (see Step 2) fine-tunes the generator-related weight parameters of the VAE-FGAN network.

As depicted in Fig 1, encoder E converts the input measurement data into hidden variables and maps them to latent variables, which subsequently reconstructs the data through generator G. The generated samples are obtained by generator G from the learning distribution. Then, the generated data and the measurement data are mixed into the discriminator D. The discriminator utilizes the GAN structure to discriminate whether the input sample is generated or real. The generated data is used as a training dataset in the classifier, and the original data is used as a test dataset.

Fig 2 illustrates the network structure comprising the encoder E, generator G and discriminator D in the VAE-FGAN network, corresponding to the third part of Fig 1. In this network structure *k* indicates the convolutional kernel size, *c* is the channel size, and *h* denotes the number of hidden layers of the gated recurrent network (GRU). The RuLU function is chosen for both the activation function of the encoder and the generator. To optimize the discrimination performance of the discriminator, the activation function of the discriminator is different from those of other convolutional layers; thus, the LeakyReLU function is adopted here.

### 3.2. Variational autoendocer generative adversarial network

VAE and GAN are two classic generative models. While GAN generates high-quality data, reaching the Nash equilibrium during adversarial training is challenging, leading to mode collapse, where the generator produces unreasonable outputs. Conversely, variance measures the loss function of VAE training, which can result in generated data that lacks fidelity to the original dataset's features. However, the latent vectors it learns during decoding can be used to generate the original data. To fully utilize the advantageous features of the two major generative models, we designed a highly stable VAE-GAN model capable of learning features. Like the traditional VAE, it learns the feature distribution of the input data through encoding and reconstruction. During the training process, the encoder E extracts and compresses the sample features in the complete dataset and encodes them into the latent space z through a linear network, where z is the important data feature information being latently captured.The generated new samples are provided by generator G based on the description of latent variable z, and $p(z)$ is the standard normal distribution of the coding space. By applying variational inference, G keeps making posterior distribution $q(z|x)$ closer to the expected distribution and chooses KL scatter as part of the loss function to calculate the distance between the two distributions [31]. The similarity function is shown in Equation (1).

$$L = -E_{q(z|x)}[log p(x|z)] + D_{KL}(q(z|x) \| p(z))$$

(1)

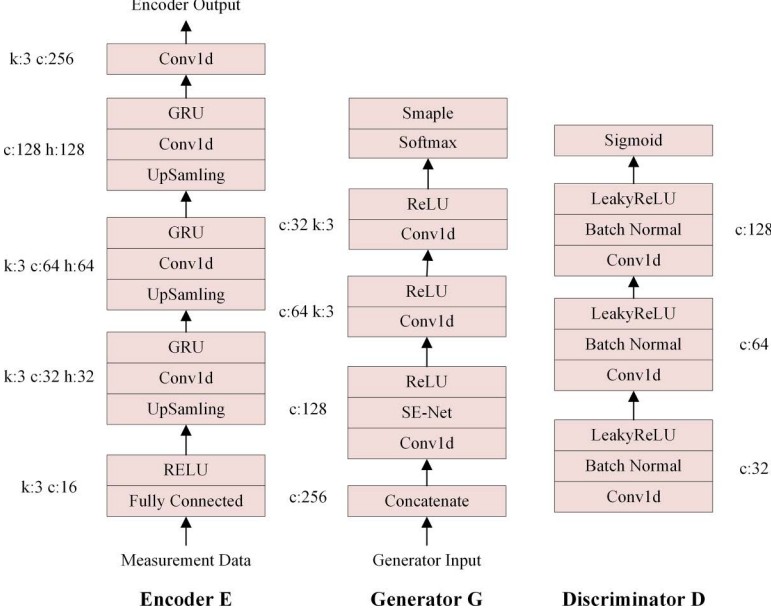

**Fig 2. Network structures of the encoder, decoder, and discriminator.**

In this network, encoder E and discriminator D form a GAN, and the encoder is considered as generator G. Unlike traditional GAN sampling from noise, our generator G generates samples similar to the training samples by capturing the distribution of the training samples. While determining accuracy of the generated data, the discriminator also classifies the data [32]. The loss function is shown in Equation (2).

$$L_{GAN} = \log(D(x)) + log(1 - D(G(z)))$$
(2)

The unique network structure of the VAE-GAN enables our generator to extract deep features of the data obtained by the encoder. The encoder bridges $z$ and real sample $x$, thereby giving the discriminator the ability to recognize different features of the data. Combining the advantages of two traditional networks not only ensures the stability of the model, but also ensures the quality of the generated data. In this way, we can extract deeper data features.

### 3.3. Design of the GRU module

As discussed previously, VAE-GAN significantly improves training stability and the quality of generated data compared to the original GAN. As a result, we incorporated the GRU module into encoder E based on the VAE-GAN model, improving encoder E's ability to acquire the data's deep semantics and the quality of the generated data.

A GRU module is introduced to encoder $E$ to improve its capability of obtaining the deep semantic information of the data. The GRU is a variant RNN similar to LSTM and has superior results in processing natural language and data prediction [33]. The interpolation of missing data in the present paper is analogous to the work of GRU data prediction, where the data information is preserved at different moments by updating and resetting gates [34]. As shown in Fig 3, three quantities are specified for our GRU module: output $h_{t-1}$ of the previous moment, input $x_t$ of the current moment, and output $h_t$ of the current moment. $z_t$ and $r_t$ in Equations (3) and (4) are the update gate and reset gate, respectively.

$$z_t = \sigma\left(W_z \cdot [h_{t-1}, x_t]\right)$$
(3)

$$r_t = \sigma\left(W_z \cdot [h_{t-1}, x_t]\right)$$
(4)

The GRU possesses unique advantages in learning data features. Considering the characteristics of the train dataset used in this paper, a new encoder structure is designed, as shown in Fig 4. The ability of the GRU to learn using contextual data is explored to extract deep inter-data features. Here, the first layer data is used as the output $h_{t-1}$ of the previous step of the GRU module. The second layer data is used as input $x_t$ of the current step of the GRU module. The useful data information is reserved and $h_t$ is output. Then, output data $h_t$, data from first layer and second layer are fused with the feature semantics and subsequently output to the next GRU module. This structural design highly integrates the correlation between data and, thereby fully combining the underlying feature information.

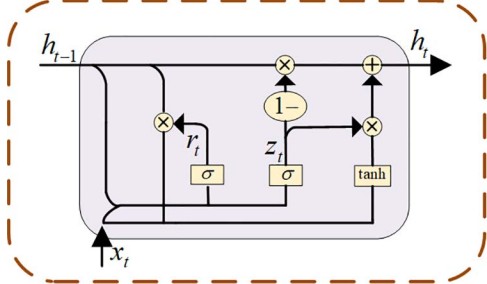

**Fig 3. GRU module.**

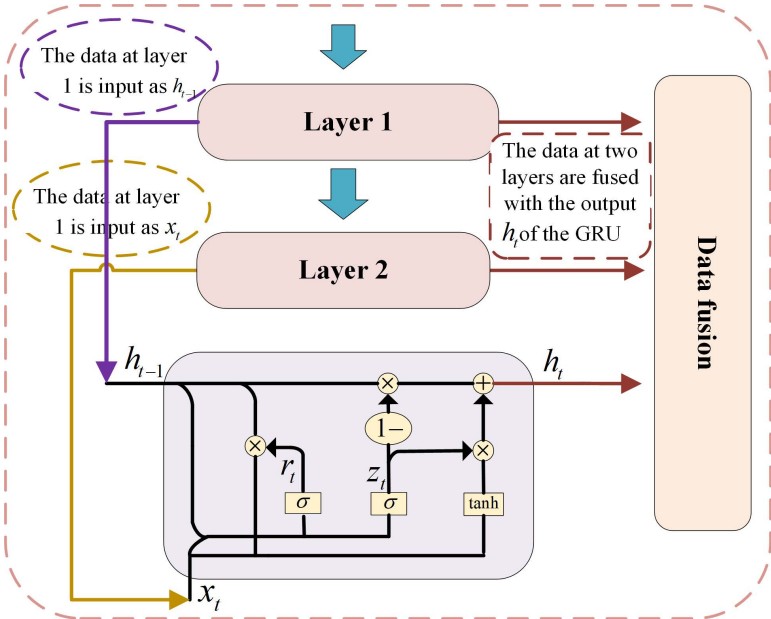

**Fig 4. Design and application of the GRU module.**

### 3.4. SE-NET attention mechanism

It is of great importance to design a high-performance encoder E and generator G for learning deep features of data and generating high-quality data. Based on the encoder structure designed in the previous section, an SE-NET attention mechanism module is added to the encoder and generator separately. A weight is used to indicate the importance of each channel in the next stage.

As shown in Fig 5, the SE-NET attention mechanism mainly consists of an SE module, a squeeze operation, an excitation operation, and feature fusion. After assigning weights to each channel and performing the squeeze operation, the network obtains a global description. The excitation operation and feature fusion enable the fully connected layer to fuse feature information [35].

It can be seen from Fig 5 that this attention mechanism executes in three steps. First, squeezing is done by globally pooling and compressing the output encoding of the previous layer into a statistical vector, which is calculated as

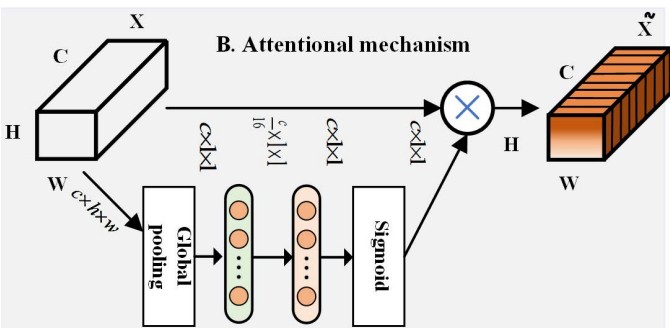

**Fig 5. SE-NET attention mechanism.**

 

$$z_t = F_{sq}(h_t) = \frac{1}{k} \sum_{t=1}^{k} h_i^{(t)}$$

(5)

where $h = [h_1, h_2 \cdots h_i]$ is the output encoding of the upper layer, and $z = [z_1, z_{2,} \cdots z_i]$ is the statistical vector after compression transformation. Since the effect of using average pooling is significantly better than applying maximum pooling, we obtain $z$ using average pooling.

Second, excitation involves the fusion of all input features through a fully connected layer of two nonlinear activation functions while a Sigmoid function keeps the output ultimately within a specific interval. This is formally expressed as

$$A = F_{ex}(Z) = \sigma_2 \left( W_2 \sigma_1 \left( W_1 Z \right) \right)$$

(6)

where $\sigma_1$ and $\sigma_2$ denote the ReLU and Sigmoid functions, respectively. By scaling the values of $W_1, W_2$ and functions, the number of model parameters are reduced and computational speed is increased.

Third, a final fusion process is undertaken to fuse the channel weights obtained from the previous two operations with the original features and then assign weights to their features by a simple multiplication operation.

## 3.5. Measurement of missing data reconstruction

Because the trained VAE-GAN model theoretically generates data that can satisfy the pattern of the quantitative data distribution, it is necessary to find the most reasonable value to replace the missing part among the countless data generated. Inspired by the reconstruction of missing images, the train's measured data samples generated by the VAE-GAN model should satisfy authenticity as well as contextual constraints. For this purpose, we adopt a suitable method for the interpolation of measured data, in which a two-part loss function is defined to evaluate the applicability of interpolation.

First, the feature data containing the missing samples need to be described appropriately. Therefore, we create a binary mask matrix M. For the missing part of the data sample, the value of the corresponding mask matrix M part is 0 and the complete part is 1. We calculate the Hadamard product of measured data X and M, and the different degrees of missing data are expressed through the resulting matrix.

Second, the proposed model aims to obtain reconstructed data that is as similar as possible to the original measured data without changing the complete measured data. Therefore, a context-bounded similarity $L_r$ is defined as shown in Equation (7). The generator needs to continuously generate the data that best matches the complete data so that the reconstructed data has a consistent contextual relationship with the complete data.

$$L_r(z) = \|X \odot M - G(z) \odot M\|_2$$

(7)

where, $X$ is the measured data containing missing values, and $G(z)$ is the generated data sample. It is important to note that this formula is only a metric for reconstructed data and non-missing parts.

The discriminator loss $L_d$ can make the reconstructed data as close to the complete data as possible. This $L_d$ is defined as

$$L_d(z) = -D(G(z))$$

(8)

The loss function of the reconstructed missing measured data consists of similarity loss and discriminator loss:

$$L(z) = L_r(z) + \lambda L_d(z)$$

(9)

## 4. Experimental results

### 4.1. Algorithm preparation and dataset processing

Our experiments are implemented using Python and PyTorch. The hardware environment is an Intel(R) Xeon(R) E-2124G CPU (with 3.41 GHz frequency), and NVIDIA GeForce GTX 1660 GPU. In complicated and changing operating environments, the key to a train having a long and durable operation relies on high quality control and maintenance. In reality, train operation and maintenance data are non-linear, discrete, and time-varying. Our experiments use the equipment operation and maintenance data of a train collected over 32 days. The same equipment are targeted across the data collection period, and five major equipment features are chosen specifically: maximum and minimum DC voltages, and maximum, minimum, and average AC voltages. These features are selected to maintain a strong correlation between data features. As shown in Table 1, we obtain 1630 complete data points. To verify the interpolation effect of the proposed model on the missing data, the data is arranged in chronological order, and random sampling is used to produce missing sample data. The features maximum, minimum, and average AC voltage values are used for pre-training; meanwhile, the maximum and minimum DC voltage values are used for fine tuning and interpolating during the main stage of training. After migrating the main training model by the good parameters of the pre-training model, 100 pieces of sample data are used to fine tune the parameters and the remaining 250 pieces of sample data are randomly sampled for use as missing data. Different extents of missing data and interpolation allows us to test the generalizability of the proposed model. The data used when training the model during the experiment must be guaranteed to be complete (i.e., with no missing values). In order to evaluate the model's performance when dealing with missing data, the fine-tuned data and the validation data must not be repeated.

### 4.2. Parameter setting and evaluation indices

The pre-trained parameters and their settings are shown in Table 2.

For evaluation of missing data reconstruction, this paper applies two indicators, the mean absolute error [36] (MAE) and mean absolute error percentage [37] (MAPE), to access the performance of model reconstruction, which calculate as shown in Equations (10) and (11).

$$MAE\left(x, \hat{x}\right) = \frac{1}{n}\left(\sum_{i=1}^{n}\left|x_i - \hat{x}_i\right|\right)$$

(10)

**Table 1. Selection and division of small sample dataset.**

| Pre-training model | | | Main training model | |
|---|---|---|---|---|
| AC max. | AC min. | AC aver. | DC max. | DC min. |
| 1280 data points | | | 350 data points | |

**Table 2. Pre-trained parameters.**

| Parameter | Value |
|---|---|
| learning rate of pre-trained encoder E | 0.001 |
| learning rate of discriminator D | 0.001 |
| learning rate of generator G | 0.0002 |
| learning rate of the main training model | 0.0002 |
| BATCH_SIZE | 1280, with 100 Epoch cycles |

$$MAPE(x, \hat{x}) = \frac{\sum_{i=1}^{n} \left| \frac{x_i - \hat{x}_i}{x_i} \right|}{n} \times 100$$

(11)

where, $x_i$ is the original data, and $\hat{x}_i$ is the complete data after reconstruction. The changes in these two indicators represent model performance on missing data reconstruction.

## 4.3. Missing data reconstruction quality

As mentioned above, we employed complete datasets of train operation and maintenance data in our experiments. We use a binary mask matrix with the complete data for the Hadamard product calculation, which allows us to represent the missing data. Considering the uncertainty and uncontrollability of the location of missing measured data during the actual operation of a train, the generated mask matrix is set randomly, where 1 represents complete data and 0 represents missing data. The quantity of missing measured data is controlled by controlling the quantity of 0s in the mask matrix. For a 20% missing rate of 250 data points, the quantity of missing data points in the generated mask matrix fluctuates slightly above and below 50.

As shown in Fig 6, assuming that the measurements in the train system numbered 2, 5, 8, 9, and 14 are all lost due to communication failure, based on the contextual feature relationship between the data and the a priori knowledge of the data, the proposed model can reconstruct the data as shown Fig 6 (see the blue dots). It can be seen that, for specific missing data, the proposed VAE-FGAN can obtain reconstruction results very close to normal values thanks to the contextual feature relationship of the measured data.

After randomly removing data, we proceeded with reconstruction. The MAE and MAPE values of the reconstruction results were calculated from comparing the measured data and the reconstructed data (Table 3). When the missing rate (i.e., the proportion of data that is removed) is ≤0.2, MAE and MAPE have low values and do not change dramatically, and the model has good reconstruction accuracy. However, when the missing rate is >0.4, the two indicators increase quickly. Although increasing the amount of missing data affects the reconstruction performance, the overall error between the reconstructed and original data is small. The proposed VAE-GAN demonstrates high reconstruction accuracy even when the train operation and maintenance data is missing at a large scale.

Table 4 compares our proposed model with traditional methods of missing data reconstruction, such as KNN and EM algorithms. These conventional methods exhibited lower accuracy due to failing to account for missing points' location and feature types. We also compared our approach with the method in Literature 10, which applied the more commonly used WGAN model to reconstruct the missing data. While this method yielded higher accuracy than traditional techniques, it was specifically designed for regular, temporal data with dominant features, and thus, its performance on our dataset was limited. The method's accuracy in Literature 11 makes our proposed model seem short, as our targeted feature extraction network significantly enhances reconstruction accuracy.

As a complete measured data group is available for this study, and the data missing is created randomly by controlling the mask matrix, there is a small difference between the MAE value and MAPE values of each experiment. After several experiments, the fluctuation between MAE and MAPE was derived. As shown in Table 5. We conducted ablation comparison experiments to ensure that the model can achieve optimal performance. Our findings indicated that using only the VAE or GAN networks alone did not yield superior reconstruction accuracy. After combining the two models into the VAE-GAN, the learning of the discrete data features significantly improved. When adding the SE-NET attention mechanism module directly to the VAE-GAN network model, the accuracy was improved, but not significantly, probably because of the few data classes in this experiment, small network layers, and insignificant impact of the attention mechanism on channel weighting. The reconstruction accuracy saw a significant enhancement with the implementation of the GRU fusion module, and the benefits of the attention mechanism increased as the number of network layers rose following the integration of the GRU module.

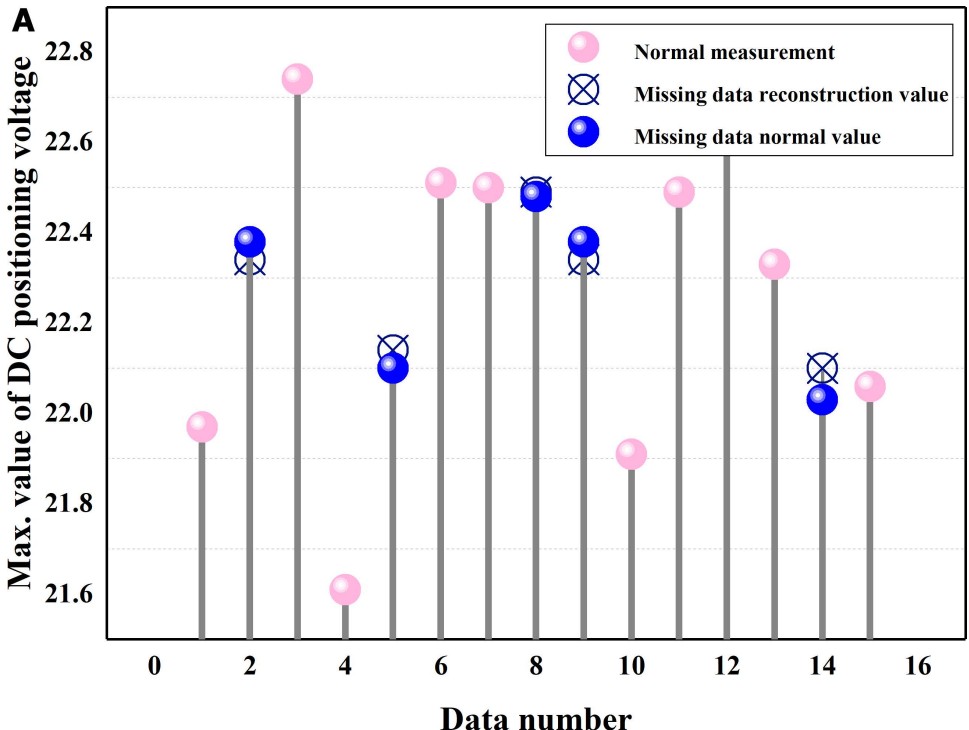

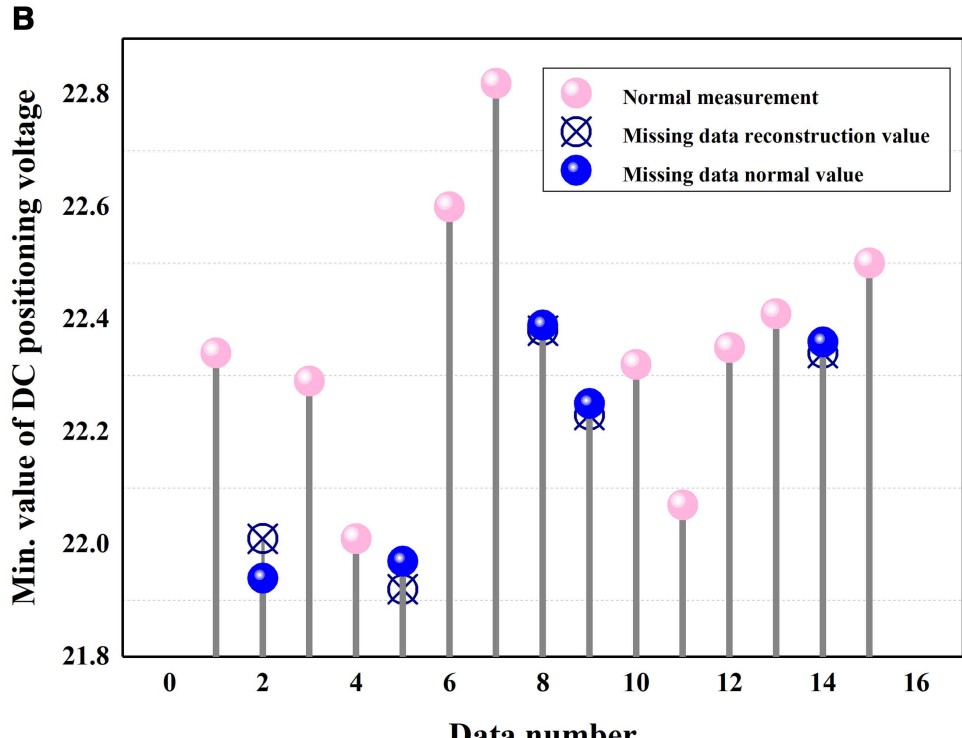

**Fig 6. Reconstruction results for specific missing data.**

**Table 3. Missing data reconstruction validation using indicators.**

| Missing rate | Maximum DC voltage value | | Minimum DC voltage value | |
|---|---|---|---|---|
| | MAE | MAPE | MAE | MAPE |
| 0.1 | 0.0206 | 0.3089 | 0.1824 | 0.7541 |
| 0.2 | 0.0314 | 0.3505 | 0.1623 | 0.6659 |
| 0.3 | 0.0462 | 0.4154 | 0.1991 | 0.8164 |
| 0.4 | 0.1341 | 0.8951 | 0.3989 | 1.2415 |
| 0.5 | 0.2198 | 1.1187 | 0.5846 | 1.4710 |

**Table 4. Comparative experiments of different methods.**

| Model | Maximum DC voltage value | | Minimum DC voltage value | |
|---|---|---|---|---|
| | MAE | MAPE | MAE | MAPE |
| KNN | 3.2851 | 3.1854 | 2.5841 | 2.2146 |
| EM | 3.1165 | 2.9501 | 2.6584 | 2.0189 |
| Literature [14] | 0.1010 | 0.9564 | 0.3123 | 1.6161 |
| Literature [18] | 0.0782 | 0.6769 | 0.2746 | 0.9166 |
| VAE-FGAN | 0.0314 ± 0.005 | 0.3505 ± 0.03 | 0.1623 ± 0.01 | 0.6659 ± 0.03 |

**Table 5. Comparison of missing data reconstruction using different models.**

| Model | Maximum DC voltage value | | Minimum DC voltage value | |
|---|---|---|---|---|
| | MAE | MAPE | MAE | MAPE |
| GAN | 0.1016 | 1.2674 | 0.3394 | 2.2842 |
| VAE | 0.0976 | 1.2234 | 0.2132 | 1.2561 |
| VAE-GAN | 0.0846 | 1.1891 | 0.2098 | 0.9263 |
| VAE-GAN +SE-NET | 0.0808 | 1.1010 | 0.1901 | 0.9261 |
| VAE-GAN +GRU | 0.0444 | 0.573 | 0.1678 | 0.7132 |
| VAE-FGAN | 0.0314 ± 0.005 | 0.3505 ± 0.03 | 0.1623 ± 0.01 | 0.6659 ± 0.03 |

The reconstruction error-to-zero ratio is the ratio of the reconstructed data that is the same as the original data. As shown in Fig 7, when the missing rate is < 50%, the reconstruction error-to-zero ratio curve shows no obvious trend of descending, which means a large part of the reconstructed data matches the original data. Meanwhile, when the missing rate is > 50%, an noticeable decreasing trend appears; but the difference between the reconstructed and generated data is not still noticeable, which indicates that the proposed VAE-FGAN can adequately learn the potential features of the measured data and work well with specific small data samples; most importantly, VAE-FGAN can reconstruct missing data without errors in contextual information.

In train operation scenarios, measured data at certain moments may be missing; long-term data may also go missing, such as maintenance data of train roof equipment. Therefore, it is necessary to consider whether the generated measured data meets the characteristics of the contextual data distribution. The reconstruction quality of the measured data can be evaluated by interpolating the generated data to the original data.

Our proposed model reconstructs missing data for small data samples of selected DC voltage by learning the characteristics of selected AC voltage data. The reconstruction quality is shown in Fig 8. For the 250 sample data points taken for each feature, 50 are randomly chosen and treated as missing. The orange curve represents the distribution of the

**A**

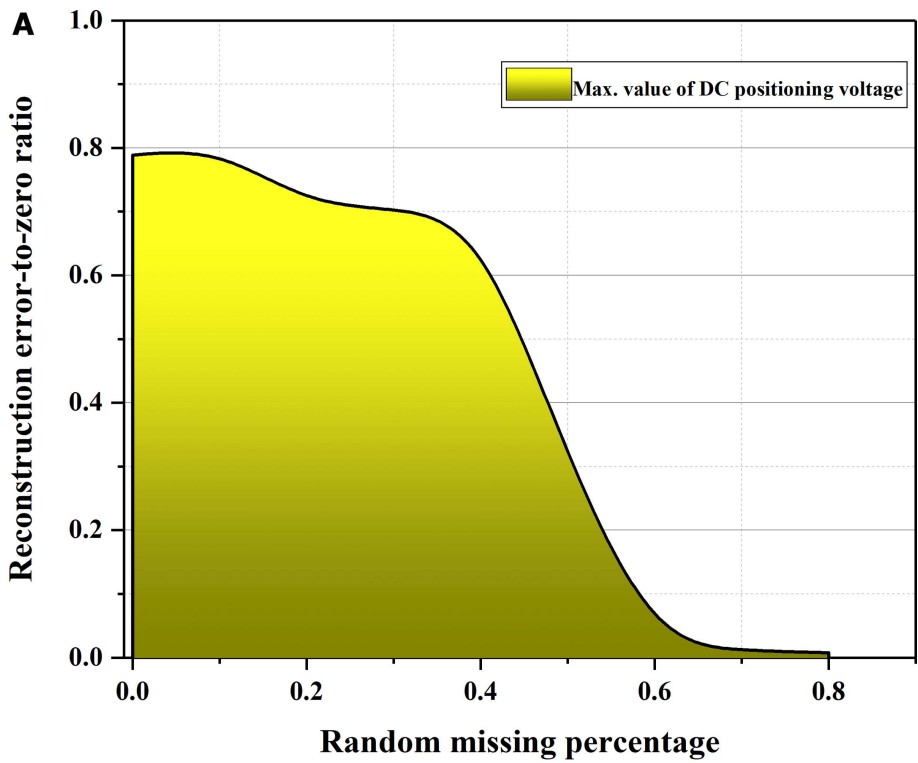

**B**

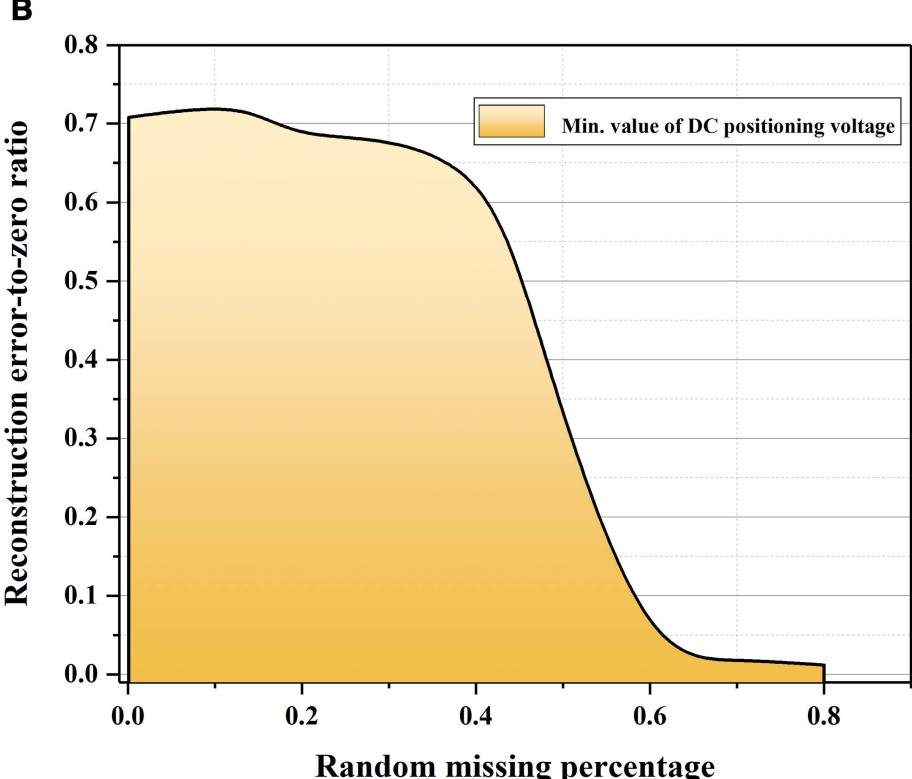

**Fig 7. Reconstruction error-to-zero ratio.**

**A**

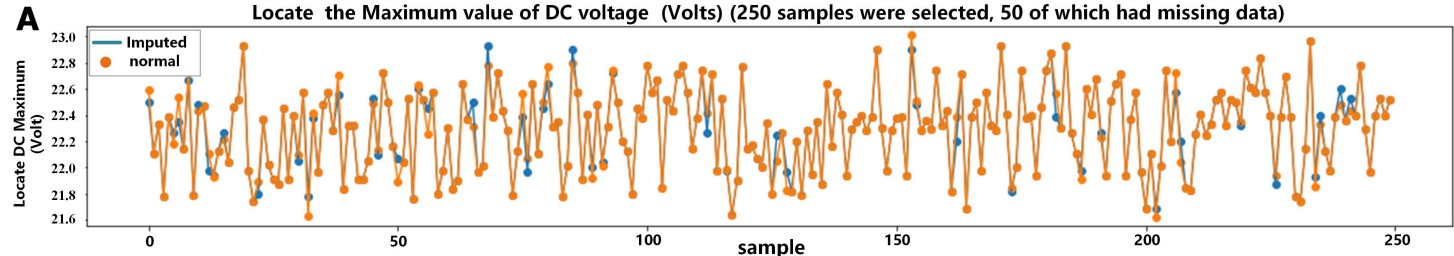

**B**

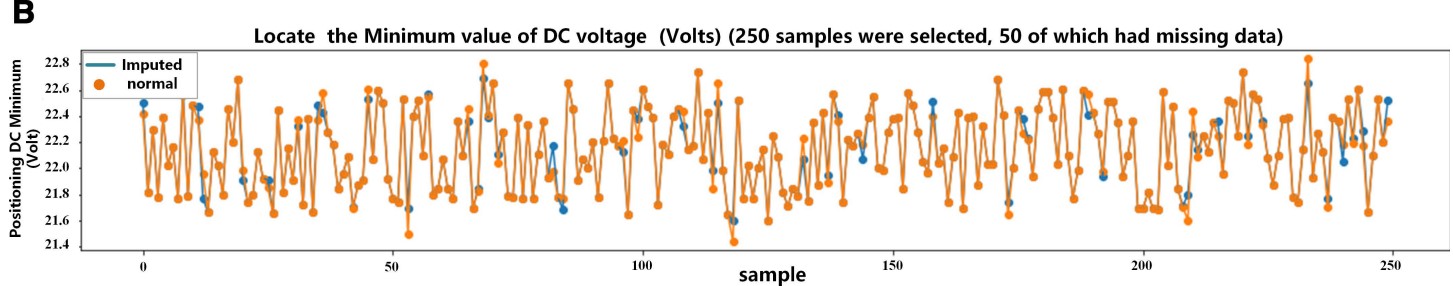

**Fig 8. Visualization of data reconstruction quality.**

original data while the blue curve represents the reconstructed data. The degree of overlap of the two curves reflects the difference between these two kinds of data. It can be seen that our VAE-GAN can greatly restore the original data information distribution with high reconstruction accuracy by learning the feature patterns of the data obtained from the same equipment.

## 5. Conclusions and future work

This paper proposes a new variational autoencoder semantic fusion generative adversarial network, which provides a missing data reconstruction strategy for addressing problems with small data samples through introducing migration learning. First, a VAE is introduced to replace the generator in the original GAN to eliminate the instability of the generation process caused by the adoption of random noise. A GRU module is then added to this encoder to semantically fuse the underlying features of the data with the higher-level features.

Second, an SE-NET attention mechanism is introduced throughout the generative network to enhance the expression of data features through feature extraction. Finally, parameter sharing is achieved through migration learning and model pre-training, thus eliminating the difficulty in training the model due to the small amount of operation and maintenance data. Experimental results show that the reconstruction quality indicators MAE and MAPE are kept below 1.5 with our model. Moreover, our model outperforms existing models, and the distribution characteristics of the reconstructed data complies with the distribution patterns of the measured data.

To achieve inter-data feature learning, further work can be carried out on constructing the coupling relationship between spatio-temporal data for multi-source data in different spatio-temporal modalities. Data reconstruction with domain migration and different datasets should also be studied.

## Supporting information

**S1 Data.**

(ZIP)

## Author contributions

**Conceptualization:** Jing He.

**Methodology:** Zhenwen Sheng.

**Writing – review & editing:** Hongrun Chen.

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
