## [Decision Letter · Decision Letter 0]

3 Jan 2025

PONE-D-24-51993Reconstruction of Missing Data in Transferred Generative Adversarial Networks with Small Sample DataPLOS ONE

Dear Dr. Chen,

Thank you for submitting your manuscript to PLOS ONE. After careful consideration, we feel that it has merit but does not fully meet PLOS ONE’s publication criteria as it currently stands. Therefore, we invite you to submit a revised version of the manuscript that addresses the points raised during the review process.

We look forward to receiving your revised manuscript.

Kind regards,

Zeeshan Ahmad

Academic Editor

PLOS ONE

Reviewers' comments:

Reviewer's Responses to Questions

**Comments to the Author**

1. Is the manuscript technically sound, and do the data support the conclusions?

Reviewer #1: Yes

Reviewer #2: Yes

2. Has the statistical analysis been performed appropriately and rigorously? 

Reviewer #1: Yes

Reviewer #2: Yes

3. Have the authors made all data underlying the findings in their manuscript fully available?

Reviewer #1: No

Reviewer #2: No

4. Is the manuscript presented in an intelligible fashion and written in standard English?

Reviewer #1: Yes

Reviewer #2: No

5. Review Comments to the Author

Reviewer #1: Under special working conditions, data collection systems of heavy-duty trains may be faced with a small sample size and missing data when executing measurement, operation, or maintenance tasks. The authors set up a frame of migration learning generative adversarial network for small data samples, in which a new variational autoencoder semantic fusion generative adversarial network (VAE-FGAN) is developed to reconstruct missing data. The topic is interesting. Some comments are given as follows:

1. The motivation for using GRU and VAE needs to be provided. There are various methods available for use.

2. The introduction needs to be more detailed. The author should summarize the existing work and provide specific advantages and disadvantages.

3. The description of Figure 1 needs to be more detailed, highlighting the main contributions.

4. How were the results of Figure 2 obtained?

5. The experimental results are not sufficient. More comparative methods need to be included.

Reviewer #2: This paper tackles the important challenge of reconstructing missing data in small sample scenarios, a common issue in data collection systems for heavy-duty trains operating under special working conditions. The authors propose an innovative migration learning framework for generative adversarial networks (GANs) specifically designed to handle small data samples. The introduction of a novel variational autoencoder semantic fusion GAN (VAE-FGAN) is a significant contribution, showing promise in effectively reconstructing missing data. However, while the paper demonstrates technical merit, there are areas that require further attention and refinement. Below are my detailed comments and suggestions:

1. To enhance the paper’s formatting consistency and professionalism, the authors should ensure that all instances of “et al.” are italicized throughout the manuscript. Furthermore, the Introduction section would benefit from a brief overview of the subsequent sections. Including a sentence such as, “The remainder of this paper is organized as follows. Section 2 introduces… Section 3 discusses…” will provide the reader with a clear roadmap and improve the overall readability of the paper.

2. The manuscript lacks a dedicated discussion on related work, which is crucial for contextualizing the contribution within the broader field. It is strongly recommended to add a new section titled “Related Work” immediately following the Introduction. This section should include an overview of the basic theory of GANs and their various applications. Key papers that should be discussed include: “COutfitGAN: Learning to Synthesize Compatible Outfits Supervised by Silhouette Masks and Fashion Styles” and “Learning to Synthesize Compatible Fashion Items Using Semantic Alignment and Collocation Classification: An Outfit Generation Framework”. These papers exemplify GAN applications in other fields and can serve as a comparative basis to highlight the novelty of the authors’ approach. Additionally, a review of existing work on GANs applied to missing data reconstruction is necessary to provide a comprehensive background and demonstrate how the proposed method advances the state of the art.

3. The title of Section 2.1 is overly lengthy and could be simplified for clarity and conciseness. A succinct title would not only improve readability but also align better with standard academic writing practices.

4. Fig. 4 lacks clarity regarding the model’s inputs and outputs, which are critical for understanding the workflow. It is recommended that the authors refine the figure by explicitly labeling these elements or adding a brief explanatory note. This improvement will significantly enhance the figure’s utility for readers.

5. There are typographical errors in the manuscript that need to be corrected. For example, “RuLU” should be replaced with the correct term “ReLU.” Additionally, Fig. 2 should be carefully reviewed for similar errors or inconsistencies to ensure the accuracy and professionalism of the presentation.

6. PLOS authors have the option to publish the peer review history of their article (what does this mean? ). If published, this will include your full peer review and any attached files.

**Do you want your identity to be public for this peer review?** For information about this choice, including consent withdrawal, please see our Privacy Policy .

Reviewer #1: No

Reviewer #2: No

---

## [Author Response · Author response to Decision Letter 0]

12 Mar 2025

Reviewer 1

1. The motivation for using GRU and VAE needs to be provided. There are various methods available for use.

Response: Thanks for your suggestions and comments. We have incorporated the motivation of using VAE and GRU in Sections 3.2 and 3.3. VAE and GAN are two classic generative models. GAN generates good-quality data, but it is challenging to reach the Nash equilibrium during adversarial training, and it will collapse during the process, leading to the generator “cheating” the generator. Moreover, the generated data is very unreasonable. The variance measures the loss function of VAE training, which generates data that does not preserve the data features. Moreover, the latent vectors it learns during decoding can be used to generate the original data. As a result, we designed a highly stable VAE-GAN model capable of learning features to leverage the benefits of the two major generative models. Like the traditional VAE, this model learns the feature distribution of the input data through encoding and reconstruction. During the training process, the encoder E extracts and compresses the sample features in the complete dataset and encodes them into the latent space z through a linear network, where z is the important data features information being latently captured.

VAE-GAN dramatically improves training stability and the quality of generated data compared to the original GAN. Therefore, we incorporated the GRU module to encoder E based on the VAE-GAN model, improving encoder E’s ability to acquire the data’s deep semantics and the quality of the generated data.

2. The Introduction needs to be more detailed. The author should summarize the existing work and provide specific advantages and disadvantages.

Response: Many thanks for your suggestions and comments. We have modified the Introduction section: First, traditional missing data reconstruction methods, such as EM and KNN, fall short in describing the correlation between complex data features and different devices by reconstructing the missing data with purely mathematical models. Second, the more popular GAN models can solve the data missing problem effectively. However, due to the particularity of our dataset, it is challenging to train these GAN models with discrete data. Thus, there is no guarantee that these models can generate samples that match the original data distribution when starting from random background noise, which can lead to the vanishing of gradients and hinder reaching Nash equilibrium. While some variants of GAN models have been developed and show improved accuracy with increasing layers, they also place higher demands on the dataset and require a larger volume of training data. As a result, their effectiveness is limited when applied to small sample datasets. Considering the potential of transfer learning in small sample scenarios, we are inspired to create a data reconstruction architecture that leverages transfer learning techniques.

3. The description of Figure 1 needs to be more detailed, highlighting the main contributions.

Response: Many thanks for your suggestion. In Section 3.1, we have described Figure 1 in more detail to highlight our innovation. Moreover, in the first paragraph of part 2, we have added a brief overview of the subsequent sections to provide a roadmap for the reader.

4. How were the results of Figure 2 obtained?

Response: Figure 2 depicts the network structure of encoder E, generator G, and discriminator D in the VAE-FGAN network, which is the third part of Figure 1. This figure visualizes the network design. We have also modified this part in Section 3.1.

5. The experimental results are not sufficient. More comparative methods need to be included.

Response: We have incorporated relevant comparative experiments in Table 4 to compare our proposed model with the traditional and some methods in the literature.

Reviewer 2

1. To enhance the paper’s formatting consistency and professionalism, the authors should ensure that all instances of “et al.” are italicized throughout the manuscript. Furthermore, the Introduction section would benefit from a brief overview of the subsequent sections. Including a sentence such as, “The remainder of this paper is organized as follows. Section 2 introduces… Section 3 discusses…” will provide the reader with a clear roadmap and improve the overall readability of the paper.

Response: Many thanks for your suggestions. We have reformatted the entire paper to make it more professional and improved its readability by including its roadmap at the end of the first section and the beginning of the third section.

2. The manuscript lacks a dedicated discussion on related work, which is crucial for contextualizing the contribution within the broader field. It is strongly recommended that a new section titled “Related Work” be added immediately following the introduction. This section should include an overview of the basic theory of GANs and their various applications. Key papers that should be discussed include: “COutfitGAN: Learning to Synthesize Compatible Outfits Supervised by Silhouette Masks and Fashion Styles” and “Learning to Synthesize Compatible Fashion Items Using Semantic Alignment and Collocation Classification: An Outfit Generation Framework”. These papers exemplify GAN applications in other fields and can serve as a comparative basis to highlight the novelty of the authors’ approach. Additionally, a review of existing work on GANs applied to missing data reconstruction is necessary to provide a comprehensive background and demonstrate how the proposed method advances the state of the art.

Response: A section devoted to related work has been incorporated into the revised version.

3. The title of Section 2.1 is overly lengthy and could be simplified for clarity and conciseness. A succinct title would not only improve readability but also align better with standard academic writing practices.

Response: We have shortened the title of Section 3.1(corresponding to the original manuscript as Section 2.1) to improve readability.

4. Fig. 4 lacks clarity regarding the model’s inputs and outputs, which are critical for understanding the workflow. It is recommended that the authors refine the figure by explicitly labeling these elements or adding a brief explanatory note. This improvement will significantly enhance the figure’s utility for readers.

Response: Thank you for the suggestions. We have used lines of different colors for inputs and outputs in Figure 4 and added short notes.

5. There are typographical errors in the manuscript that need to be corrected. For example, “RuLU” should be replaced with the correct term “ReLU.” Additionally, Fig. 2 should be carefully reviewed for similar errors or inconsistencies to ensure the accuracy and professionalism of the presentation.

Response: We have corrected some errors in Figure 2, making our paper more accurate and professional!

---

## [Decision Letter · Decision Letter 1]

19 Mar 2025

Reconstruction of Missing Data in Transferred Generative Adversarial Networks with Small Sample Data

PONE-D-24-51993R1

Dear Dr. Chen,

We’re pleased to inform you that your manuscript has been judged scientifically suitable for publication and will be formally accepted for publication once it meets all outstanding technical requirements.

Kind regards,

Zeeshan Ahmad

Academic Editor

PLOS ONE

Additional Editor Comments (optional):

Reviewers' comments:

Reviewer's Responses to Questions

**Comments to the Author**

Reviewer #1: (No Response)

Reviewer #2: All comments have been addressed

2. Is the manuscript technically sound, and do the data support the conclusions?

Reviewer #1: (No Response)

Reviewer #2: Yes

3. Has the statistical analysis been performed appropriately and rigorously? 

Reviewer #1: (No Response)

Reviewer #2: Yes

4. Have the authors made all data underlying the findings in their manuscript fully available?

Reviewer #1: (No Response)

Reviewer #2: Yes

5. Is the manuscript presented in an intelligible fashion and written in standard English?

Reviewer #1: (No Response)

Reviewer #2: Yes

6. Review Comments to the Author

Reviewer #1: (No Response)

Reviewer #2: The revisions made to this paper have significantly enhanced its quality. The authors have addressed previous concerns effectively, and the manuscript now demonstrates clarity, coherence, and rigor. I have no further comments and consider the paper ready for publication.

7. PLOS authors have the option to publish the peer review history of their article (what does this mean? ). If published, this will include your full peer review and any attached files.

**Do you want your identity to be public for this peer review?** For information about this choice, including consent withdrawal, please see our Privacy Policy .

Reviewer #1: No

Reviewer #2: No

---

## [Editor Report · Acceptance letter]

PONE-D-24-51993R1

PLOS ONE

Dear Dr. Chen,

I'm pleased to inform you that your manuscript has been deemed suitable for publication in PLOS ONE. Congratulations! Your manuscript is now being handed over to our production team.

Kind regards,

on behalf of

Dr. Zeeshan Ahmad

Academic Editor

PLOS ONE